# Higher income individuals are more generous when local economic inequality is high

**Joel H. Suss** [ID] *

London School of Economics & Political Science and Bank of England, London, United Kingdom

* j.suss@lse.ac.uk

## Abstract

There is ongoing debate about whether the relationship between income and pro-social behaviour depends on economic inequality. Studies investigating this question differ in their conclusions but are consistent in measuring inequality at aggregated geographic levels (i.e. at the state, region, or country-level). I hypothesise that local, more immediate manifestations of inequality are important for driving pro-social behaviour, and test the interaction between income and inequality at a much finer geographical resolution than previous studies. I first analyse the charitable giving of US households using ZIP-code level measures of inequality and data on tax deductible charitable donations reported to the IRS. I then examine whether the results generalise using a large-scale UK household survey and neighbourhood-level inequality measures. In both samples I find robust evidence of a significant interaction effect, albeit in the opposite direction as that which has been previously postulated–higher income individuals behave more pro-socially rather than less when local inequality is high.

**Data Availability Statement:** US data is available publicly. Data on charitable giving available from IRS website: https://www.irs.gov/statistics/soi-tax-stats-individual-income-tax-statistics-zip-code-data-soi. Inequality data and information used for

## 1 Introduction

Does economic inequality moderate the relationship between income and pro-social behaviour? This question has become increasingly important as inequality has risen sharply over the last half-century in the US, UK and many other countries [1,2]. Existing research situates the potential effect of inequality on pro-social behaviour within a larger, and largely inconclusive, debate on whether the rich are more or less generous than the poor [3–7]. A notable paper finds that the relationship between income and pro-social behaviour depends on inequality, with a negative interaction between inequality at the US state-level and individual income [8]. However, more recent studies, using data on inequality at the state, region and country-level find no significant interaction effect [9] and the opposite sign [10,11] when looking at charitable giving and volunteering behaviour. Follow-up analysis using new survey data [12] has failed to provide clarity [13].

Understanding whether and how inequality moderates the relationship between income and pro-social behaviour is critical from a societal perspective. If higher inequality reduces pro-social behaviour among the rich, for example reducing the amount of charitable giving, rising inequality might reinforce itself in the absence of countervailing forces [14]. In

control variables available from Census Bureau and downloaded from NHGIS: https://data2.nhgis.org/main UK inequality data has been calculated by me for a separate publication (https://journals.sagepub.com/doi/abs/10.1177/0308518X231154255) and is available for download on my site here: https://github.com/jhsuss/uk-local-inequality. UK survey data is secure access and therefore cannot be shared publicly, although access can be requested from: http://doi.org/10.5255/UKDA-SN-6614-14.

**Funding:** The author received no specific funding for this work.

**Competing interests:** The author has declared that no competing interests exist.

aggregate, charitable donations by individuals were estimated at over $300bn in the United States in 2019 according to Giving USA, with a large proportion of the total going to education, poverty alleviation, and other organisations which aid in reducing inequality. To put that figure into perspective, the amount the US government spends on poverty alleviation has been estimated at just under $400bn for 2018 [15].

Scholars investigating the link between inequality and pro-social behaviour generally assume that macro-level inequality is the appropriate spatial unit of analysis, however there is good reason to believe it is not. Importantly, it is not clear whether individuals receive the macro-level inequality 'treatment' for two reasons. First, there is wide variation of local inequality within macro-level areas–for example, individuals in San Francisco and Sacramento, or London and Liverpool don't experience the same level of inequality and generally live in very different contexts. Inequality, as measured by the Gini coefficient on incomes, iss 11% larger in San Francisco than Sacramento (using 2018 county-level data; [16]), and 17% larger in London relative to Liverpool based on the Gini coefficient of housing values [17]. Using a single figure to represent multiple local contexts obscures variability in lived experiences. Fig 1 depicts this wide variation for the US by plotting income inequality at the state-level and by ZIP-code, sub-state areas that are better approximations of local communities than states [18,19].

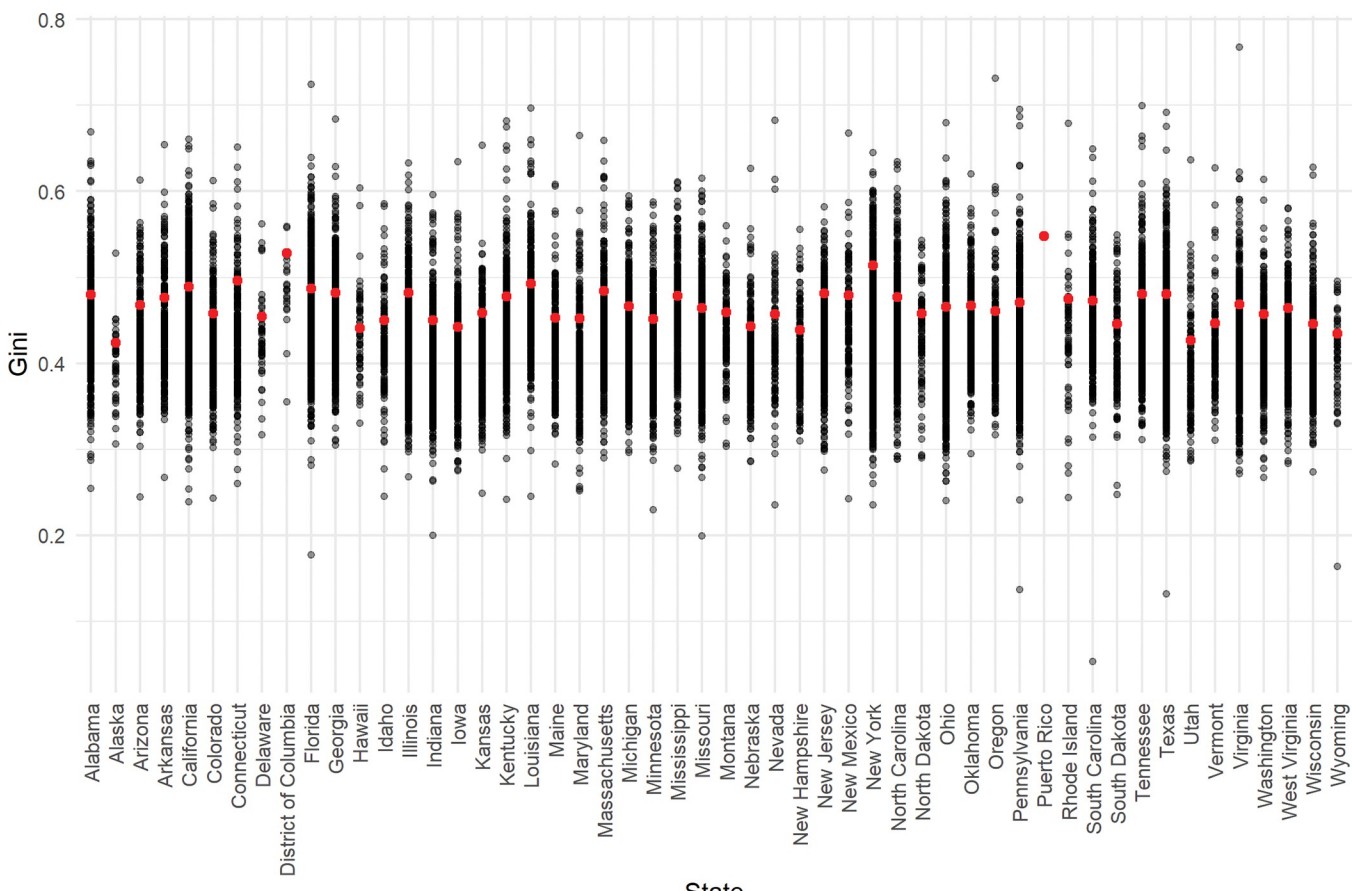

**Fig 1. US State and ZIP-level income inequality, 2014–18.** Note: Fig 1 shows US state-level income inequality (red dot) and within-state ZIP-level income inequality (black dots). Inequality is measured using the Gini coefficient of incomes and obtained from the American Community Survey, 2014–2018 [16].

Second, a nascent body of work calls into question whether actual levels of national inequality are accurately perceived. Indeed, perceptions tend to be far off from reality [20–23]. This is important because perceptions of inequality rather than actual levels have been found to be more relevant for attitudes and behaviour [24–26]. On the other hand, local measures of economic inequality have been shown to be associated with subjective perceptions of national inequality [24,27,28]. This is because local contexts serve as important sources of information used by individuals to make sense of wider society [29]. This is not only the case for distributional perceptions–in other domains as well, individuals utilise local information when forming judgments about macro variables, e.g. national economic performance or unemployment [30–32]. Thus, if national assessments of inequality matter for pro-social behaviour, local inequality is likely to be an important factor through its influence on perceptions of national inequality.

From a theoretical perspective, greater inequality is thought to increase the social distance between economic classes [8,14], thereby attenuating support for poverty alleviation through charitable giving or redistributive policies on the part of the rich. In so far as rising aggregate inequality goes hand in hand with increasing residential segregation [33], this should indeed result in greater social distance through the logic of greater physical distance. Less contact which arises through residential segregation along economic lines might allow negative stereotypes and stigmatisation to grow [34,35]. However, the social distance mechanism would suggest that greater inequality within local areas, where rich and poor have greater opportunity for interactions, could work instead to reduce social distance through greater inter-group contact [36–38]. This contact could take the form of, for example, simple observations of those in relative need or through more economically heterogeneous friendship networks, helping to reduce stigmatisation and increase empathy [29,39]. Experienced from a local perspective, economic inequality might therefore have a positive moderating impact, increasing the pro-social behaviour of the rich.

This paper tests this hypothesis using granular data on local economic inequality–measured by the Gini coefficient–for the US and UK. I match this with information on income and charitable giving from tax returns in the US (N = 165,621) and a large household survey in the UK (N = 39,289) for the period spanning 2016–2018. I focus on these countries and time period due to the availability of large-scale and detailed information on income, charitable giving (used as the main proxy for pro-sociality) and, crucially, spatially-granular estimates of economic inequality which are not generally available for other countries.

Across both samples I find evidence of a positive moderating effect of local inequality on the relationship between income and charitable giving. For example, in the US sample, the highest income group (over $200k in gross income) sees an expected 15 percentage point increase in the propensity to donate relative to the lowest income category ($25k or under) when ZIP-code area inequality increases by one standard deviation. In the UK sample, higher local inequality is associated with larger percentages of individual income donated to charity (i.e. the main effect of inequality on giving is positive), and this effect is even larger for individuals with higher incomes relative to lower incomes.

These results generalise to a second pro-social behaviour–volunteering, and are robust to varying the definition of inequality and the spatial unit of analysis. Moreover, in order to mitigate endogeneity concerns stemming from selection effects–i.e. the possibility that higher-income, more pro-social individuals choose to live in areas of higher inequality–I restrict the UK sample to only those who live within a short distance (5 mile radius) of where they grew up (N = 3,258). The effect size grows larger when analysing this much smaller sample for both the intensive and extensive margins of charitable giving (i.e. the percent donated conditional on donation, and propensity to donate to charity).

In what follows, I first provide some background literature and formulate a hypothesis on the moderating effect of local inequality (Section 2). Section 3 details the data and methodology, Section 4 presents the results, and Section 5 concludes with a discussion of limitations and areas for further research.

## 2 Background

### 2.1 Income and pro-sociality

There is a large body of work examining how socio-economic status influences pro-social behaviour. A number of influential studies provide evidence, in the form of laboratory and naturalistic experiments, that higher class individuals tend to be relatively less pro-social, for example donating less [3] or behaving more unethically [4]. Other studies document that the negative relationship between class and pro-sociality even extends to children [40], with family incomes affecting the pro-sociality of their kids [41]. The mechanism for this relationship is argued to be that poorer individuals, who face relatively greater difficulties in life and the need to share resources with others, tend to be more egalitarian and compassionate [42,43], whereas richer individuals tend to be more individualistic and narcissistic [44,45].

Other studies, however, reach the opposite conclusion, finding that higher income individuals tend to be more pro-social. For example, better off individuals are more likely to return mis-delivered letters [7,46] and display generosity in real-world or experimental settings [47]. Part of the ambiguity in these findings might be due to different national contexts, for example due to variation in social norms or religiosity across countries [48], types of pro-social behaviour studied, or due to other contextual factors, such as the visibility of pro-sociality and reputational concerns [49,50]. A large-scale study testing the link between socio-economic status and pro-sociality across different contexts and behaviours finds a predominately positive relationship, with higher income individuals more likely to donate to charity and donate a larger proportion of income for example [5].

### 2.2 Economic inequality as a moderator?

With this backdrop, a notable paper reasoned that economic inequality might be an important contextual variable that moderates the effect of income on pro-sociality [8]. The authors find evidence across both observational and experimental studies that macro-level inequality (measured at the US state-level) interacts negatively with income. The authors rationalise their findings by suggesting that inequality increases the social distance between rich and poor, with the former group developing a greater sense of entitlement where inequality is high. However, other studies, examining the moderating effect of inequality across numerous countries and regions, find either positive or null effects [9–11].

In each of these studies, the spatial level of analysis is an important feature which is not critically assessed. Indeed, one potential reason why the results are ambiguous is due to the aggregated nature of the chosen spatial unit (state, region or country), which, as noted in the introduction, is unlikely to proxy for how inequality is experienced, objectively or subjectively.

There is also nascent body of work on the main effect of economic inequality for pro-social behaviour [14,51–53]. These studies, too, reach conflicting conclusions and operationalise inequality either at an aggregate level or as part of an artificial lab manipulation. One exception is the study by [54], in which the authors find that changes in income inequality at the level of Canadian urban areas and local communities is associated with increases in charitable giving.

Taken together, this suggests the need to measure inequality at a more disaggregated level, one that better approximates the relevant spatial unit for experiencing inequality and forming assessments of aggregate inequality [24,27,28].

## 2.3 Local inequality and social contact

Theoretically, local inequality might also be relevant as a moderator in and of itself. Local environments are where individuals come into contact with others, forming social networks and accumulating knowledge through social observation [29,55,56]. From this perspective, a local setting that is relatively unequal, where rich and poor live side by side, might work to reduce the social distance between economic classes [57]. This view is supported by inter-group contact theory [36–38], which, while originally focused on how and under what conditions inter-ethnic group contact would reduce prejudice, has been extended in further directions to encompass the implications of local economic contexts [58]. However, it is important to make clear here the assumption that inequality in local settings necessarily involves rich and poor living side by side, i.e. that they are forced in local contexts to be in contact with one another, whereas inequality in more aggregated settings may involve economic segregation to a degree that means rich and poor do not typically come in contact with or observe one another. Of course there may also be neighbourhoods that are completely economically segregated, where social contact is minimal, and therefore local economic segregation is likely to be an important factor to consider alongside local inequality. I address this in the empirical analysis below by including a control for local economic segregation (available for the UK sample).

Social contact suggests that local inequality might lead relatively rich individuals to act more pro-socially. This leads to the formulation of the following *local inequality hypothesis*: local inequality is expected to interact positively with income for pro-social behaviour. In other words, the relationship between income and pro-social behaviour should depend on local inequality such that higher income households will be relatively more generous.

While the focus of the literature and the hypothesis here is on the pro-sociality of the relatively well-off, the behaviour of the poor is also important. Here, there is also suggestive evidence consistent of a positive interaction effect: local inequality might trigger 'class consciousness' [59] or feelings of relative deprivation [60,61] on the part of the relatively disadvantaged. Local inequality might therefore lead to reductions in pro-sociality on the part of the poor, particularly in situations where pro-social behaviours are directed at cross-class counterparts [62].

## 3 Materials and methods

To evaluate the interaction between local inequality and income on pro-social behaviour, I first analyse tax data in the US on income and charitable donations at the ZIP-level, and then follow this up with a nationally representative sample of UK households, which contains questions on recent charitable giving and granular, neighborhood-level geographical markers. Across both these samples, I focus on two outcome measures: 1) the amount donated to charity conditional on having donated something (i.e. the intensive margin), and 2) the likelihood of donating at all (i.e. the extensive margin).

### 3.1 US study

For the US, I analyse charitable donations from aggregated and anonymised tax returns provided by the US Internal Revenue Service for 2018 [63]. The data is disaggregated at various levels of geography. In the main analysis, I focus on the most granular disaggregation–the ZIP-code level–but also examine alternative spatial units (counties and states) to test the local inequality hypothesis, with counties considered to be (relatively imperfect) approximations of local settings and states too large. ZIP-code areas have a mean population of 14,041 (SD = 15,846).

The IRS data is broken down by adjusted gross income bracket, i.e. households are grouped by income bracket. There are six income groups in total across all ZIP-code areas (N = 165,621): $1-$24,999; $25,000-$49,999; $50,000-$74,999; $75,000-$99,000; $100,000-$199,999; $200,000+). Thus, I analyse variation in the extensive and intensive margins of charitable giving by income group (modeled as a categorical variable) and ZIP-code. In particular, I take the percentage of households in each income group that donate some money to charity (the extensive margin; M = 9.3%), and the average percent of income donated conditional on some households donating (the intensive margin; M = 1.17%, SD = 1.21 percentage points). I obtain income inequality estimates, taken as the Gini coefficient in the baseline analysis (top 5% share of income in robustness checks), from the American Community Survey (ACS) for the five year period 2014–2018 [16]. As indicated in Fig 1, ZIP-level inequality is highly dispersed (M = 0.43, SD = 0.06, Max. = 0.77, Min. = 0.02).

To estimate the interaction effect, I use multilevel linear modeling (via the lme4 package in R; [64]), with income groups nested within ZIP-code areas. I take the logistic functional form for the percentage of returns donating per income group since it varies between 0 and 1 (with the number of returns per group as weights), and I logarithmise the percent of income donated conditional on at least some households in the income group donating since the values are strictly positive and the distribution is skewed rightward. (Using simple linear models rather than taking the logistic functional form or logarithm of the dependent variable does not affect the results (unreported).) I include state fixed effects and control for a number of ZIP-level variables which are likely to be relevant: median income, population density (residents / square mile), the proportion of the population that is White, has a university degree or above, is below the poverty line, is young (under 18 years old), and old (65 years old and over). S1 Table in S1 File document provides descriptive statistics.

In the main analysis, I drop ZIP-code areas with less than 1,000 residents to mitigate concerns around inequality estimates for smaller areas. In robustness checks, I further restrict the sample according to size of the ZIP-code area population (above and below a threshold of 50k residents).

While there are advantages to using administrative tax data over surveys, in particular the data covers all tax returns, there are also some important downsides. The income groups provided by IRS data are top coded and coarse, with households subsumed into income groups. The analysis might therefore be prone to the ecological fallacy [65,66], whereby associations observed at the group level are different from that of individuals. Finally, poorer households are more likely to take the standard deduction rather than itemise charitable donations [67], which means that charitable giving is likely to be under-reported for lower income groups. While there is no reason to suspect that this itemisation differential by income group varies by level of inequality, I cannot rule this out.

## 3.2 UK study

To provide further evidence regarding the relationship between income, local inequality and pro-social behaviour, and to verify that the results generalise outside the US, I turn to a large-scale survey of UK households matched with neighborhood-level inequality measures. As with the US sample, local inequality is taken as the Gini coefficient in the main analysis, but in robustness checks I check whether the results hold when inequality is measured as the top 1% share. I combine information on self-reported charitable giving in Wave 8 of Understanding Society (N = 39,289; gathered in 2016–17; [68]) with data on economic inequality at the neighborhood-level. Due to a lack of granular data on income or wealth (unlike the US, the UK does not gather this information as part of the decennial census), the UK inequality measures are based on housing values for around 23 million UK addresses [17].

Neighborhoods are taken as the UK Middle Lower Super Output Area (MSOA)–census areas that are population weighted (M = 7,787, SD = 1,600) and meant to adhere to natural neighborhood boundaries. As with the US sample, in robustness checks I vary the spatial unit of analysis, going even more granular (Lower Super Output Areas; LSOAs) and more aggregated (Local Authority Districts; LADs). LSOAs are constituent building blocks of MSOAs (population M = 1,471, SD = 428), and LADs are larger local government boundaries (population M = 161,138, SD = 109,066). Unlike LSOAs and MSOAs, LADs are not population weighted and are more akin in size variation to US counties. Varying the spatial unit of analysis mitigates concerns that arise from the Modifiable Areal Unit problem [66] and ensures that the results are not sensitive to the decision around which spatial boundaries to use.

Understanding Society asks respondents whether they gave money to charitable organisations in the last 12 months (M = 0.66) and, if so, how much was given (M = £241, SD = £623). Survey respondents also report their gross monthly income (annualised M = £48,750, SD = £34,954) and other information that allows me to account for important individual differences. I control for age, gender, education, ethnicity, political orientation (proxied by reported political party support), religiosity, and marital status. I also control for neighbourhood-level factors: median house prices, economic segregation (measured using the Multigroup Entropy Index [69,70] with Ouput Areas–building blocks of MSOAs–as the sub-units), and population density, and I include fixed effects for UK region (see S2 Table in S1 File for full descriptive statistics). As with the US data, I use multilevel modelling with individual respondents nested within MSOAs to test the interaction between income and MSOA-level inequality.

I also check whether these results generalise to other pro-social behaviours using the UK survey. In particular, I test whether inequality interacts with income when self-reported volunteering behaviour is the outcome measure. Once more, I look at both the extensive margin (whether volunteered in the last 12 months; M = 0.18) and intensive margins (hours spent volunteering in the last 4 weeks; M = 11.17, SD = 18.98). Lastly, in order to mitigate endogeneity concerns, I restrict the UK sample to those who have never moved. This reduces the sample dramatically (N = 3,258), but alleviates selection concerns, e.g. more generous and richer individuals choosing to live in areas that are also more unequal.

## 4 Results

### 4.1 Study 1: Local inequality and charitable giving in the United States

Table 1 provides the estimated standardised coefficients for the moderating effect of local inequality on income for charitable donations. Column 1 (Column 4) are the results without any controls for the extensive (intensive) margin, Column 2 (Column 5) includes controls and only the main effect of income and inequality, and Column 3 (Column 6) introduces the interaction between them. There are significant and generally positive interaction effects between inequality and income for both the extensive and intensive margins when comparing the lowest income group (the omitted category; less than $25k) with the highest income group ($200k or more). In particular, the coefficients on the interaction terms for the highest income group are 0.145 and 0.141 for the extensive and intensive margins respectively (Column 3 and 6 of Table 1). This means that a one standard deviation increase in ZIP-code area inequality (i.e. an increase in the Gini by 0.055) is associated with a 15.6 percentage point increase in the likelihood of donating and a 14.1 percentage point increase in the average amount donated for the highest income group relative to the lowest income group, holding all other variables constant.

Note that the main effect of inequality for the extensive margin is positive, which suggests an increase in inequality by one standard deviation increases the likelihood of donating for the lowest income group by 16.5%. The main effect of inequality for the intensive margin is

**Table 1. Regression results for US ZIP-level inequality.**

| | Dependent variable: | | | | | |
|---|---|---|---|---|---|---|
| | **Percent of income group donating to charity** | | | **Average percent of income donated** | | |
| | **(1)** | **(2)** | **(3)** | **(4)** | **(5)** | **(6)** |
| Gini | 0.204*** | 0.172*** | 0.153*** | -0.020*** | 0.021*** | -0.042*** |
| | -0.01 | -0.009 | -0.009 | -0.006 | -0.005 | -0.006 |
| $37,500 | 1.368*** | 1.365*** | 1.368*** | 0.468*** | 0.465*** | 0.463*** |
| | -0.002 | -0.002 | -0.002 | -0.006 | -0.006 | -0.006 |
| $62,500 | 2.331*** | 2.329*** | 2.331*** | 0.869*** | 0.871*** | 0.867*** |
| | -0.002 | -0.002 | -0.002 | -0.005 | -0.006 | -0.006 |
| $87,500 | 2.759*** | 2.760*** | 2.758*** | 1.033*** | 1.037*** | 1.034*** |
| | -0.002 | -0.002 | -0.002 | -0.006 | -0.006 | -0.006 |
| $150,000 | 3.337*** | 3.340*** | 3.337*** | 1.355*** | 1.356*** | 1.353*** |
| | -0.002 | -0.002 | -0.002 | -0.005 | -0.005 | -0.005 |
| $200,000+ | 4.275*** | 4.318*** | 4.275*** | 1.899*** | 1.898*** | 1.892*** |
| | -0.002 | -0.002 | -0.002 | -0.006 | -0.006 | -0.006 |
| Gini:$37,500 | -0.033*** | | -0.033*** | 0.021*** | | 0.018*** |
| | -0.002 | | -0.002 | -0.006 | | -0.006 |
| Gini:$62,500 | -0.054*** | | -0.055*** | 0.045*** | | 0.042*** |
| | -0.002 | | -0.002 | -0.006 | | -0.006 |
| Gini:$87,500 | 0.005*** | | 0.005*** | 0.083*** | | 0.081*** |
| | -0.002 | | -0.002 | -0.006 | | -0.006 |
| Gini:$150,000 | 0.036*** | | 0.036*** | 0.080*** | | 0.077*** |
| | -0.002 | | -0.002 | -0.006 | | -0.006 |
| Gini:$200,000+ | 0.145*** | | 0.145*** | 0.138*** | | 0.141*** |
| | -0.002 | | -0.002 | -0.006 | | -0.006 |
| Median income | | 0.131*** | 0.135*** | | 0.175*** | 0.176*** |
| | | -0.014 | -0.014 | | -0.008 | -0.008 |
| Ln(Population) | | -0.032*** | -0.034*** | | -0.043*** | -0.042*** |
| | | -0.007 | -0.007 | | -0.004 | -0.004 |
| White (%) | | -0.362*** | -0.366*** | | -0.140*** | -0.140*** |
| | | -0.008 | -0.008 | | -0.005 | -0.005 |
| Poor (%) | | -0.264*** | -0.260*** | | 0.030*** | 0.030*** |
| | | -0.011 | -0.011 | | -0.006 | -0.006 |
| 25+ with college degree (%) | | 0.363*** | 0.360*** | | -0.108*** | -0.108*** |
| | | -0.012 | -0.012 | | -0.007 | -0.007 |
| Age 65+ (%) | | 0.0001 | 0.001 | | 0.181*** | 0.183*** |
| | | -0.008 | -0.008 | | -0.004 | -0.004 |
| Age less than 18 (%) | | 0.001 | 0.003 | | 0.019*** | 0.020*** |
| | | -0.009 | -0.009 | | -0.005 | -0.005 |
| Constant | -5.415*** | -5.076*** | -5.073*** | -5.685*** | -5.230*** | -5.225*** |
| | -0.01 | -0.042 | -0.042 | -0.006 | -0.023 | -0.023 |
| State fixed effect | N | Y | Y | N | Y | Y |
| Random effect level | ZIP | ZIP | ZIP | ZIP | ZIP | ZIP |
| Observations | 128,183 | 128,101 | 128,101 | 78,745 | 78,732 | 78,732 |
| Log Likelihood | -1,044,169.00 | -1,054,956.00 | -1,036,765.00 | -58,136.28 | -54,588.60 | -54,167.19 |

*(Continued)*

**Table 1.** (Continued)

| Akaike Inf. Crit. | 2,088,365.00 | 2,110,041.00 | 2,073,670.00 | 116,300.50 | 109,309.20 | 108,476.40 |

Note

*p<0.1

**p<0.05

***p<0.01.

Continuous independent variables are standardised (i.e. mean is grand-centred, standard deviation = 1). Standard errors are in parentheses. The income group reference category is $1-$24,999. Inequality data from American Community Survey, charitable donation and income group from IRS.

negative ($\exp(\beta) = 0.959$, $p < 0.01$), and while the coefficients for the second and third lowest income groups are positive, the net effect is such that an increase in local inequality is associated with a reduction or no change in the average amount donated. Thus, local inequality here appears to increase the generosity of the better off and reduce it for lower income groups, consistent with the discussion on relative deprivation and 'class awareness' above.

To better interpret the moderating effect of local inequality, I plot the expected extensive and intensive margin per income group as inequality is varied (5th to 95th percentile)–see Fig 2. The figure demonstrates that the effect of income on giving depends on inequality, with higher income groups generally donating more (aside from the lowest three income groups for the intensive margin). These findings contrast with those of [8]. Indeed, the effect is in the opposite direction, with the relationship between income and giving depending positively on the level of local inequality, both for the extensive and intensive margins.

In order to verify the robustness of these results, I run a number of checks. First, I alter the measure of inequality used. Rather than the Gini coefficient, I take the top 5% share of total income. Also, rather than using measures of inequality for 2014–2018, I use the Gini coefficients for each ZIP-code from 2007–2011 (the first 5-year window provided by the ACS at the ZIP-level). Inequality is moderately persistent over time ($r = 0.62$). Second, I examine subsamples of the data, running regressions for small ZIP-code areas (which I define as less than 50k total population) and large ZIP-code areas (over 50k) separately. All these robustness checks generally find the same pattern of results for the moderating effect of local inequality on the relationship between income group and charitable giving (see S3 Table in S1 File).

I also examine whether the same interaction effect exists when income is interacted with county and state-level inequality, replicating the spatial unit utilised by [8,9] in the case of the latter. At this level of aggregation, the income grouping information provided by the IRS includes additional bands, including households with no income which I remove. In contrast to Table 1, the interaction between income and state-level inequality is negative and significant for the extensive margin–i.e. increased inequality reduces giving by all income groups relative to the lowest income group (incomes less than $10k) and insignificant for the intensive margin–see S4 Table in S1 File. County-level inequality moderates the relationship similar to the ZIP-code level, albeit only for the extensive margin. This underlines the importance of the chosen spatial unit of analysis and suggests micro and macro-level inequality exerts opposing forces on pro-social behaviour. Inequality measured at an aggregated level (i.e. US states) reaches the opposite conclusion or largely fails to detect an effect when there is one at the more localised and contextually relevant ZIP-level.

## 4.2 Study 2: Local inequality and charitable giving in the United Kingdom

Table 2 provides the standardised coefficient estimates for the extensive and intensive margins of charitable giving in the UK sample. Column 1 (Column 4) are the results without any

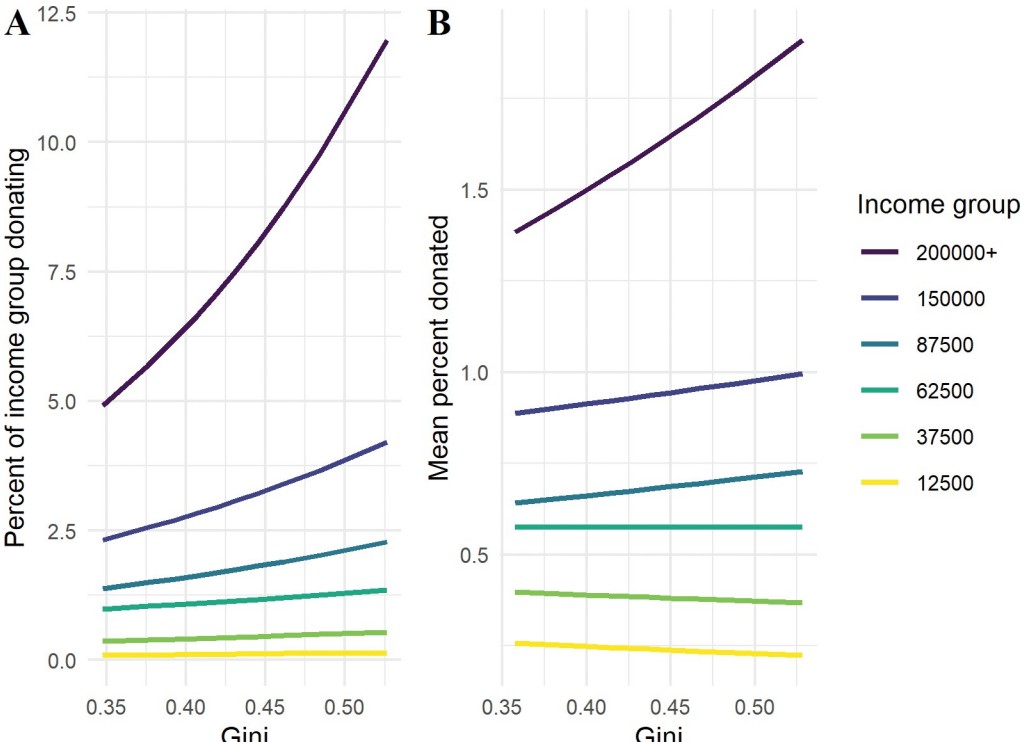

**Fig 2. Interaction between income group and ZIP-level inequality for charitable giving.** Note: Fig 2 shows the fitted values for the percent of households in each income group donating some amount (the extensive margin), and the average percent of income donated conditional on some households donating (the intensive margin) as inequality is varied (5th to 95th percentile values). All control variables are fixed at the global median value and the ZIP code is selected as 10001, New York.

controls for the extensive (intensive) margin, Column 2 (Column 5) includes controls and only the main effect of income and inequality, and Column 3 (Column 6) introduces the interaction between them. The results indicate that there is a significant interaction between income and inequality for the amount given conditional on giving ($\exp(\beta) = 1.018$, $p < 0.01$), but not for the likelihood of giving ($\exp(\beta) = 0.998$, $p > 0.1$). To better interpret the results, I once more plot the expected percent of charitable donations for those that reported donating in the previous 12 months as inequality is varied (5th to 95th percentile) for different points in the income distribution–see Fig 3.

The main effect of inequality on propensity to donate is positive and significant for both, suggesting that inequality increases the likelihood of donating ($\exp(\beta) = 1.011$, $p < .01$) and the percent donated conditional on donating ($\exp(\beta) = 1.079$, $p < .01$), irrespective of income. The sign on the income variable is negative for the intensive margin, suggesting that those with higher gross annual incomes tend to donate a lower percentage of income to charity, although the absolute amount is always increasing with income–see S5 Table in S1 File.

As with the US sample, I verify that these results are robust in a number of ways. First, I check whether these results hold for alternative measure of inequality–rather than the Gini coefficient, I take the top 1% share measure. Second, rather than taking the percent amount donated to charity, I instead use the absolute amount as the outcome measure. The results are consistent–see S5 and S6 Tables in S1 File. Next, I examine whether the relationship exists at different spatial units of analysis. First, going more granular (LSOA-level). The main effect and interaction effect remain significant and positive for the percent donated, and only the main

**Table 2. Regression results for UK MSOA-level inequality.**

| | Dependent variable: | | | | | |
|---|---|---|---|---|---|---|
| | Donated to charity? | | | Percent donated to charity | | |
| | **(1)** | **(2)** | **(3)** | **(4)** | **(5)** | **(6)** |
| Gini | 0.039*** | 0.012*** | 0.012*** | 0.139*** | 0.083*** | 0.082*** |
| | (0.003) | (0.003) | (0.003) | (0.011) | (0.012) | (0.012) |
| Income (£/year) | 0.046*** | 0.036*** | 0.036*** | -0.400*** | -0.370*** | -0.373*** |
| | (0.003) | (0.003) | (0.003) | (0.011) | (0.011) | (0.011) |
| Gini:Income | 0.001 | | -0.002 | 0.028*** | | 0.018** |
| | (0.002) | | (0.002) | (0.009) | | (0.009) |
| Economic segregation | | 0.004*** | 0.004*** | | 0.025*** | 0.025*** |
| | | (0.0002) | (0.0002) | | (0.001) | (0.001) |
| Age | | -0.064*** | -0.064*** | | 0.078*** | 0.078*** |
| | | (0.005) | (0.005) | | (0.017) | (0.017) |
| Male | | 0.105*** | 0.105*** | | 0.431*** | 0.431*** |
| | | (0.006) | (0.006) | | (0.019) | (0.019) |
| Degree | | 0.051*** | 0.051*** | | -0.161*** | -0.164*** |
| | | (0.009) | (0.009) | | (0.033) | (0.033) |
| White | | 0.073*** | 0.073*** | | 0.132*** | 0.133*** |
| | | (0.006) | (0.006) | | (0.022) | (0.022) |
| Employed | | 0.035*** | 0.036*** | | -0.025 | -0.026 |
| | | (0.006) | (0.006) | | (0.021) | (0.021) |
| Married | | 0.068*** | 0.068*** | | 0.319*** | 0.319*** |
| | | (0.006) | (0.006) | | (0.020) | (0.020) |
| Religious | | -0.018*** | -0.018*** | | -0.060** | -0.061** |
| | | (0.007) | (0.007) | | (0.024) | (0.024) |
| Labour | | 0.017 | 0.017 | | 0.125*** | 0.125*** |
| | | (0.011) | (0.011) | | (0.037) | (0.037) |
| Liberal Democrat | | -0.072*** | -0.072*** | | -0.149*** | -0.148*** |
| | | (0.007) | (0.007) | | (0.026) | (0.026) |
| Other political party | | -0.006* | -0.006* | | -0.036*** | -0.036*** |
| | | (0.003) | (0.003) | | (0.012) | (0.012) |
| Population density | | 0.001 | 0.001 | | 0.108*** | 0.106*** |
| | | (0.004) | (0.004) | | (0.015) | (0.015) |
| Median house value | | 0.027*** | 0.027*** | | 0.089*** | 0.087*** |
| | | (0.005) | (0.005) | | (0.018) | (0.018) |
| Constant | 0.670*** | 0.365*** | 0.365*** | -1.583*** | -3.096*** | -3.095*** |
| | (0.003) | (0.021) | (0.021) | (0.011) | (0.078) | (0.078) |
| Region fixed effect | N | Y | Y | N | Y | Y |
| Random effect level | MSOA and Household | MSOA and Household | MSOA and Household | MSOA and Household | MSOA and Household | MSOA and Household |
| Observations | 32,262 | 31,390 | 31,390 | 21,256 | 20,860 | 20,860 |
| Log Likelihood | -20,876.750 | -19,045.690 | -19,050.570 | -37,254.920 | -35,078.660 | -35,080.330 |

(*Continued*)

**Table 2.** (Continued)

| | Dependent variable: | | | | | |
| | Donated to charity? | | | Percent donated to charity | | |
| | (1) | (2) | (3) | (4) | (5) | (6) |
|---|---|---|---|---|---|---|
| Akaike Inf. Crit. | 41,767.500 | 38,149.380 | 38,161.150 | 74,523.840 | 70,215.320 | 70,220.660 |

Note

*p<0.1

**p<0.05

***p<0.01.

Coefficients of continuous independent variables are standardised (i.e. mean is grand-centred, standard deviation = 1). Standard errors are in parentheses. Data on inequality from Suss (2023), survey data from Understanding Society.

inequality effect is significant for the likelihood of donating, mirroring the results at the MSOA-level. Second, for LADs, the interaction is significant for both the likelihood of donating and percent donated–see S7 Table in S1 File. Importantly, when restricting the sample to those who have never moved, the main inequality effect for the likelihood of donating and the percent donated remain statistically significant, as are the interaction effects between income and inequality–see S8 Table in S1 File).

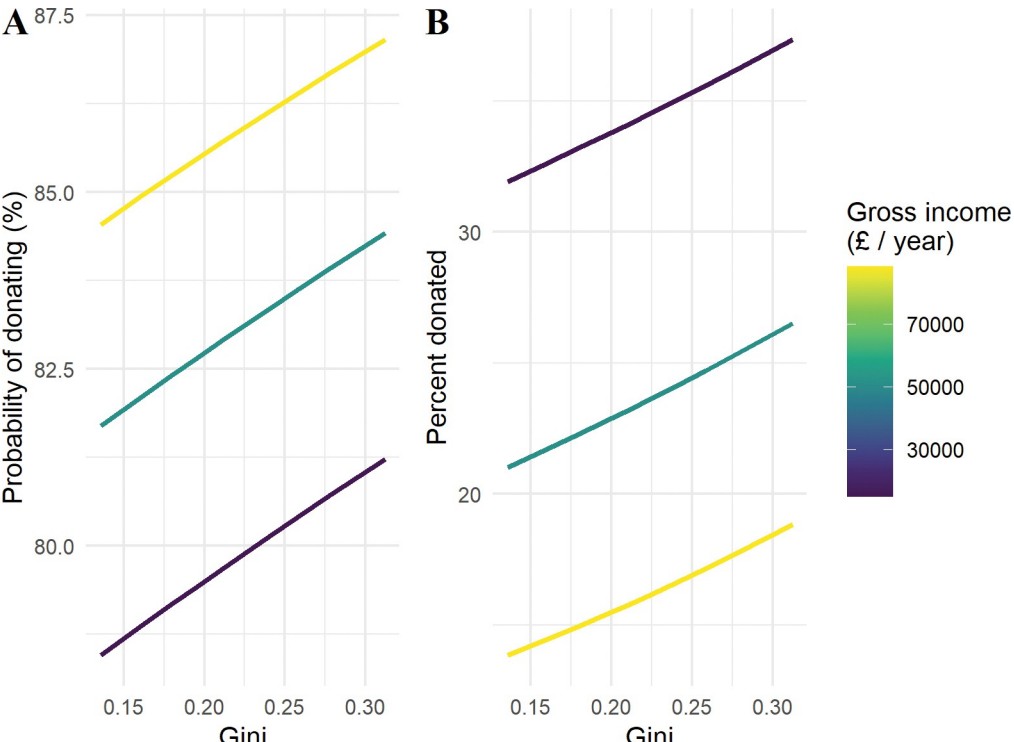

**Fig 3. Interaction between individual income and MSOA-level inequality for charitable giving.** Note: Fig 3 shows the fitted values for the probability of donating money to charity in the last 12 months (the extensive margin) and the expected percent of income donated conditional on donating (the intensive margin) by income as inequality is varied (5th to 95th percentile values). Income is set at its mean value and above and below one standard deviation from the mean. All other continuous variables are fixed at their global median value, other explanatory variables are fixed as follows: Female, White, with a degree, employed, married, religious, supporter of the Liberal Democrats, and living in London.

Looking at the moderating effect of local inequality for volunteering, I find a similar pattern of results as with charitable giving. For the hours volunteered (the intensive margin), the interaction term is positive and significant ($\beta = 0.026$, $p < 0.05$), while only the main effect of local inequality is positive and significant for the likelihood of volunteering ($\beta = 0.014$, $p < 0.01$). See S9 Table in S1 File.

## 5 Discussion and conclusion

The contribution of this paper is to examine how local rather than macro manifestations of inequality interacts with income for pro-sociality. While national inequality has been the focus of scholars, the role of local contexts, both in shaping perceptions of national inequality (e.g. by providing distributional information that is generalised to wider society) and by altering attitudes and behaviour directly (e.g. by affecting the mix of people frequently encountered and/or befriended), has been less appreciated.

The results from this analysis are clear but contrary to what has been reported in a notable paper on the moderating effect of inequality for the relationship between income and pro-social behaviour [8]–I find robust evidence for a positive interaction between income and inequality across two studies in the US and UK. Higher income individuals and households are generally more likely to donate and more generous in absolute terms when giving, and this effect is even larger in contexts of high inequality for the percent of income donated in the UK, and both the extensive and intensive margins in the US. Moreover, I find that the interaction between income and inequality generalises to volunteering.

What explains the positive interaction between income and inequality? In explaining their findings, [8] argue that higher inequality might trigger a sense of entitlement among richer individuals. The results here suggest that the opposite is the case when looking at more granular measures of inequality–local inequality is associated with greater social solidarity on the part of the rich. There is some evidence that local inequality reduces the pro-sociality of the poor, but this is confined to the intensive margin of charitable giving in the US sample. Local inequality seems to have a broadly positive effect on giving across the economic distribution in the US and UK, in line with previous findings from Canada [54].

Another possible explanation for the positive interaction, one which I am unable to explore here, does not require greater empathy on the part of the rich, just a greater sense of responsibility. The observed effect might arise from the rich wanting to give back to society as a way of satisfying their conscience [71] if they reside in local areas which is higher in inequality. Future work might seek to shed light on the mechanisms behind the positive interaction between local inequality and income for pro-sociality, perhaps by understanding how attitudes towards and reasons for charitable giving are affected by contextual economic discrepancies.

This paper is not without limitations. First, while the findings presented above are robust to different specifications and sub-sample analyses, they fall short of identifying causal relationships. The biggest concern is that the findings arise from possible selection effects, whereby individuals and households move to areas which are more aligned with their preferences with respect to inequality, and that these choices are also correlated with charitable giving. I mitigate this concern by exploiting information from Understanding Society on whether respondents live in close proximity to where they grew up. The restrictions do not change the results, but it is possible that the individuals that have not moved are those pre-disposed to the type of neighbourhood they already reside in and therefore do not have reason to move. Since I cannot rule this out, I stop short of claiming that the observed relationships are causal.

One way that scholars have tried to identify the causal effect of economic inequality on individual pro-social behaviour is by conducting experiments in the lab or online (for example,

[8,72]). However, it is unlikely that the artificial conditions and simplified representations of economic inequality within lab contexts adequately proxy for real-world experience (see also [9] who make this same argument). Moreover, economic games, such as the canonical dictator game, which scholars have typically used to study pro-social preferences experimentally, have been shown to be poorly correlated with real-world pro-social behaviour [73]. Therefore, in the absence of plausibly exogenous variation in economic inequality or a compelling instrumental variable, we have to rely on rigorously controlled observational data analysis to understand the effect of inequality on pro-social behaviour.

Future work might provide additional evidence to support the findings here. For example, while I analyse data from two separate countries, given importance of national culture and norms for pro-social behaviours [48], future work could address whether the findings here generalise to other contexts and cultures and/or time periods. Indeed, while the US and UK differ in many respects, for example in terms of political institutions and tax treatment of charitable giving, they are also be relatively similar in other ways, notably in terms of high macro-inequality levels. Moreover, the rich and poor live cheek-by-jowl in many US and UK cities like New York and London, whereas other countries might tend towards greater economic segregation in urban settings. Future work might therefore expand the set of countries examined and also consider explicitly how local segregation affects the moderating relationship of local inequality.

Future research might also investigate why macro-level and micro-level inequality may produce contradictory effects on pro-social behaviour, as documented here for US states and ZIP-areas. One possible explanation outside of the scope of this paper is that higher macro-level inequality is associated with greater economic segregation at local levels. So whereas the effect of local level inequality might be positive even in states with high inequality, higher inequality at the macro-level might mean there are simply less economically mixed communities.

Overall, this paper provides evidence that, at least at the local level, rising inequality does not reinforce itself through reduced donations to charitable organisations. Instead, local inequality gives rise to increased generosity, especially on the part of the relatively well-off.

## Supporting information

**S1 File. Supporting information.** This file provides additional tables—descriptive statistics and robustness checks.
(PDF)

## Acknowledgments

Many thanks to Ben Ansell, Paul Dolan, John Hills, Neil Lee, Kate Laffan, Jennifer Sheehy-Skeffington for valuable comments, guidance and support.

## Author Contributions

**Conceptualization:** Joel H. Suss.

**Data curation:** Joel H. Suss.

**Formal analysis:** Joel H. Suss.

**Methodology:** Joel H. Suss.

**Visualization:** Joel H. Suss.

**Writing – original draft:** Joel H. Suss.

**Writing – review & editing:** Joel H. Suss.

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
