## [Decision Letter · Decision Letter 0]

2 Jan 2023

PONE-D-22-30696High local economic inequality leads higher income individuals to be more generousPLOS ONE

Dear Dr. Suss,

Thank you for submitting your manuscript for consideration to PLOS ONE. I was fortunate to have obtained evaluations of your manuscript from three reviewers with extensive research experiences and knowledge on this topic. I have also read the manuscript carefully myself. The reviewers and I found a number of positive aspects in the research. All of us agreed that you have tackled a timely and highly relevant question regarding the association between inequality and prosocial behavior. Given the reviewers’ recommendations and the potential of your work, I am offering you the option to revise and resubmit. Although I will not reiterate the points that the reviewers have raised because their comments are clearly expressed, I will highlight some issues below.

I echo the concern raised by Reviewer 1 about the definition of inequality, which is somewhat neglected in the manuscript. The term inequality may mean different things to different people and in different contexts. I think engaging more with the definition discussion would certainly benefit the manuscript.

I also echo the serious concern raised by Reviewer 2 (see the first paragraph of the comment). It is unclear what the significance of the findings are because it seems that this research does not emphasize a theoretical advance. I struggle to find enough of a theoretical advancement and the manuscript’s theoretical contribution seems to be limited.

I also like you to account and provide more justification in the introduction for why you focused on specific cultural contexts (i.e., the US and The UK), for the selection of variables, samples, use of methods and the statistical analyses. I think that this research suffers a flaw that it is not clear whether and to what extent testing your hypotheses for people in the US and the UK was valid and meaningful. You acknowledge yourself that the UK and US differ in many respects.

You use mediation to test one of your hypothesis, but I worry you are not actively engaging with the assumptions of mediation modelling and ensuring that your analysis/model is theoretically valid (Spencer, Zanna & Fong, G. T, 2005 ; Bullock & Green, 2021; Götz et al., 2020; Stuart et al., 2022).

I also think that the structure of your manuscript at times feels quite fragmented and as Reviewer 3 points the methods appear everywhere. I recommend you to follow a standardized structure of a scientific manuscript.

We look forward to receiving the revised manuscript and thank you for considering PLOS ONE as an

outlet for your research.

Kind regards,

Milan Obaidi

Academic Editor

PLOS ONE

Journal Requirements:

Reviewers' comments:

Reviewer's Responses to Questions

**Comments to the Author**

1. Is the manuscript technically sound, and do the data support the conclusions?

Reviewer #1: Yes

Reviewer #2: Partly

Reviewer #3: Partly

2. Has the statistical analysis been performed appropriately and rigorously? 

Reviewer #1: Yes

Reviewer #2: Yes

Reviewer #3: Yes

3. Have the authors made all data underlying the findings in their manuscript fully available?

Reviewer #1: Yes

Reviewer #2: Yes

Reviewer #3: Yes

4. Is the manuscript presented in an intelligible fashion and written in standard English?

Reviewer #1: Yes

Reviewer #2: Yes

Reviewer #3: Yes

5. Review Comments to the Author

Reviewer #1: The paper studies how inequality affects prosocial behavior. Previous literature has found conflicting evidence. This paper uses tax data to calculate donations (prosocial behavior) and inequality from the ACS. The results imply that higher inequality areas are more prosocial, potentially from an increase in inter-class contact.

I really liked the paper, it is interesting and it provides an added value to the literature. I believe the paper may benefit from the following comments:

1. Inequality is not defined. You have to check the figure to realize that the Gini coefficient is used. This should be explained in the introduction and in the data section. Introduction should also include that other measures were included in the robustness section.

2. Is it possible to calculate income inequality using tax data? One challenge with the ACS is how representative is at a very disaggregated level, and the role of outliers.

3. Figure 2 should include confidence intervals. Also, the text does not explain what the estimates mean. Is it plotting marginal effects? Odd-ratios? It is not clear. The interpretation of the estimates should be clearer in the text.

4. The explanation of results in Study 1 and Study 2 seem different. Could it be possible to homogeneize a little bit more?

5. An important aspect to discuss is how preferences evolve over time. The estimates found are not fixed, but subject to culture and social norms. Then, it is possible that the relationship between inequality and prosocial behavior was modified recently, such that an important avenue for future research is how and when these preferences change.

Reviewer #2: The manuscript addresses a highly relevant topic of current debate: the consequences of income inequality on pro-social behaviour. The author even goes a step further by analysing how income inequality affects the pro-social behaviour of various income groups differently (i.e. testing the moderating effects of IE) and analyses close contacts/friendships as one potential mediator of the link between inequality and pro-social behaviour. I strongly believe that addressing these various paths and interlinkages is of high public and academic interest.

Unfortunately and in its current state, I am unable to recommend the manuscript for publication. The reasons are as follows:

1) Focus of study: The paper will benefit from a clear focus. At the moment the author seems to address various questions: the inequality-prosocial behaviour link, the moderation of inequality on the income-prosocial behaviour link, and a potential mediator (friendship). This is very appealing – but at the same time, all these questions/links have to be addressed adequately at a theoretical, conceptual and empirical level. From my perspective, this study (in its current state) does not meet these standards. However, I believe that after careful revision, this paper certainly has the potential to do so.

2) Literature review: the paper will certainly benefit from a broader overview of the literature. At the moment, the choice of the cited literature seems somewhat eclectic and its description falls short.

3) Theory: At the theoretical level, the paper is still underdeveloped and a clear specification and justification of hypotheses is missing. Adding this, will certainly improve the value of this paper. At this point, the main argument of the author focusses on the geographic unit (which is well portrayed) – but the author does not test this empirically and does not compare different unit effect. Instead, the paper tests the specific links between economic inequality, pro-social behaviour and income groups with one geographic unit (for each country: the US and the UK) – and these links are theoretically not clearly specified.

4) Measurement of Income Inequality: I fully agree with the authors’ argument that economic inequality observed at smaller geographic units, such as neighbourhood/ZIP code level, are of vital importance and even more likely to influence social behaviour – due to the mechanisms described by the author (i.e. direct experience and less biased perceptions of inequality). However, the measurement of income inequality at smaller geographic units is also more complex and more likely to be biased. As the author points out correctly, segregation effects have to be addressed – not only empirically, but also theoretically (-> if there is more segregation, people may not observe and experience as much inequality as we would expect when studying consequences of inequality at larger geographic units). I also wonder how different measures of inequality affect the result and how the author deals with the high level of correlation between inequality and other (control) variables in the analysis. Does this affect the results?

5) Data: The author uses aggregated macro data from the US as well as a combination of macro & micro data from the UK. I am not entirely sure how the results of the different studies build up on each other and can be compared. Therefore, I urge the author to carefully portray the value of each study for the study and the overall research question(s)/testing of hypotheses once they are clearly specified.

6) Methods: More information and justification is needed for the selection of variables, use of methods and the statistical analyses – this would certainly benefit the paper. For example, why is the logarithmic function of the DV preferred over a linear one (p.5). What is the number of observations for the US? Why is the mediation analysis/variable not described in section 2? Do results vary with fixed effects for region?

7) Choice of figure and tables: I advise that figures and tables should be carefully chosen to illustrate the main results. Overall, I found the tables more enlightening than most of the figures included in the paper. For tables I recommend the following: clear labelling of variables, clear specification of reference groups (for categorical variables), and the display of the constant (which is of high value for the interpretation of interaction effects).

8) Interpretation of research findings: to allow a general readership to understand the value of the results, the interpretation of interaction effects could be more carefully described. At the same time, more careful description is required with regard to the different outcome variables (US vs. UK; absolute vs. relative donation) and their comparability. I also recommend differentiating between direct and indirect effects in the mediation analysis.

9) Language: the author often uses a very technical language throughout the paper (e.g. ‘negative interaction’, ‘opposite signs’, see paragraph 1 of the introduction for illustration). This should be avoided so to make this study also attractive and understandable for a larger readership. At the same time, the paper would benefit from more precision and clarity in the use of language with regard to the conceptual linkages between inequality and prosocial behaviour and its variation with household and personal characteristics (such as income and friendships).

Reviewer #3: Reading the article causes some dissonance. On the one hand, sophisticated analyses have been conducted using multi-level modeling. But on the other hand, the description is very narrative, in the style of a sociological essay.

The generally accepted rules about the content of the various parts of the manuscript are broken. In particular, methods appear everywhere - in the introduction, in the methodological section (rightly so here) and in the presentation of results. A more formalized form of method description (data sources, samples, dependent and independent variables, statistical methods) would make it easier for a large part of the audience to understand the results.

It is also surprising that the results except for one table are included in the appendix. It is difficult for me to judge whether this material will be available only electronically. I suggest selecting key results from Study 1 and Study 2 and including at least one core table or graph in each subsection (UK and US data, respectively).

Other suggestions are as follows:

- making the aim of the study more explicit at the end of the introduction instead of a paragraph about the analyses performed

- clear distinction of the description of the sample, variables and methods of analysis in Chapter 2 (divided into Study 1 and Study2) , while limiting this type of information in the introduction and results as already highlighted above

- marking the sources of data under the tables and explaining the abbreviations

- checking the clarity of the tables (for example, what is the difference between models 1 and 2 and 3 and 4); perhaps explanatory information can be in the title

- verify that all tables and figures are assigned to US or UK populations (fig.3?).

6. PLOS authors have the option to publish the peer review history of their article (what does this mean?). If published, this will include your full peer review and any attached files.

Reviewer #1: No

Reviewer #2: No

Reviewer #3: No

---

## [Author Response · Author response to Decision Letter 0]

2 Mar 2023

Please see the attached word document "revision_reply" for my response to each comment. Many thanks to the editor and reviewers for all the time spent reviewing this work and for very helpful comments.

---

## [Decision Letter · Decision Letter 1]

12 May 2023

Higher income individuals are more generous when local economic inequality is high

PONE-D-22-30696R1

Dear Dr. Suss,

We’re pleased to inform you that your manuscript has been judged scientifically suitable for publication and will be formally accepted for publication once it meets all outstanding technical requirements.

Kind regards,

Maurizio Fiaschetti

Academic Editor

PLOS ONE

Additional Editor Comments (optional):

Reviewers' comments:

Reviewer's Responses to Questions

**Comments to the Author**

1. If the authors have adequately addressed your comments raised in a previous round of review and you feel that this manuscript is now acceptable for publication, you may indicate that here to bypass the “Comments to the Author” section, enter your conflict of interest statement in the “Confidential to Editor” section, and submit your "Accept" recommendation.

Reviewer #1: All comments have been addressed

Reviewer #3: All comments have been addressed

2. Is the manuscript technically sound, and do the data support the conclusions?

Reviewer #1: Yes

Reviewer #3: Yes

3. Has the statistical analysis been performed appropriately and rigorously? 

Reviewer #1: Yes

Reviewer #3: I Don't Know

4. Have the authors made all data underlying the findings in their manuscript fully available?

Reviewer #1: Yes

Reviewer #3: Yes

5. Is the manuscript presented in an intelligible fashion and written in standard English?

Reviewer #1: Yes

Reviewer #3: Yes

6. Review Comments to the Author

Reviewer #1: All my comments have been addressed. I have no further comments.

XXXXXXXXXXXXXXXXXXXXXXXXXXXXXXXXXX

Reviewer #3: I am very pleased to have had the opportunity to participate in the review process of this paper. The current version is much better and I fully appreciate the effort to make changes.

7. PLOS authors have the option to publish the peer review history of their article (what does this mean?). If published, this will include your full peer review and any attached files.

Reviewer #1: No

Reviewer #3: No

---

## [Editor Report · Acceptance letter]

22 May 2023

PONE-D-22-30696R1 

Higher income individuals are more generous when local economic inequality is high 

Dear Dr. Suss:

I'm pleased to inform you that your manuscript has been deemed suitable for publication in PLOS ONE. Congratulations! Your manuscript is now with our production department. 

Kind regards, 

on behalf of

Dr. Maurizio Fiaschetti 

Academic Editor

PLOS ONE